# Gut Microbiota Composition Associated with *Clostridioides difficile* Colonization and Infection

**DOI:** 10.3390/pathogens11070781

**Published:** 2022-07-08

**Authors:** Elisa Martinez, Bernard Taminiau, Cristina Rodriguez, Georges Daube

**Affiliations:** 1Laboratory of Food Microbiology, Fundamental and Applied Research for Animals & Health (FARAH), Department of Food Sciences, Faculty of Veterinary Medicine, University of Liege, 4000 Liege, Belgium; bernard.taminiau@uliege.be (B.T.); georges.daube@uliege.be (G.D.); 2Instituto de Investigación Biomédica de Málaga (IBIMA), Unidad de Gestión Clínica de Aparato Digestivo, Hospital Universitario Virgen de la Victoria, 29590 Málaga, Spain; cris.rdrz@gmail.com

**Keywords:** gut microbiota, *Clostridioides difficile*, *Clostridioides difficile* infection, asymptomatic colonization, human

## Abstract

*Clostridioides difficile* is an anaerobic Gram-positive and spore-forming bacterium. The majority of *C. difficile* strains produce two toxins, A and B, associated with the development of acute diarrhea and/or colitis. In this review, two situations are distinguished: *C. difficile* infection (CDI) and asymptomatic colonization (AC). The main objective of this review is to explore the available data related to the link between the gut microbiota and the development of CDI. The secondary aim is to provide more information on why some people colonized with toxigenic *C. difficile* develop an infection while others show no signs of disease. Several factors, such as the use of antibiotics and proton pump inhibitors, hospitalization, and age, predispose individuals to *C. difficile* colonization and/or *C. difficile* infection. The gut microbiota of people with AC showed decreased abundances of *Prevotella*, *Alistipes*, *Bacteroides*, *Bifidobacterium*, *Dorea*, *Coprococcus*, and *Roseburia*. The gut microbiota of people suffering from CDI showed reductions in the abundances of *Lachnospiraceae*, *Ruminococcaceae*, *Blautia* spp., *Prevotella* spp., *Dialister* spp., *Bifidobacterium* spp., *Roseburia* spp., *Anaerostipes* spp., *Faecalibacterium* spp. and *Coprococcus* spp., in comparison with healthy people. Furthermore, increases in the abundances of *Enterococcaceae* and *Enterococcus* were associated with *C. difficile* infection.

## 1. Introduction

*Clostridioides difficile* is an anaerobic, Gram-positive, and spore-forming bacterium recognized as the leading cause of health care-associated diarrhea. In the United States of America (USA), in 2017, the Centers for Disease Control and Prevention (CDC) considered *C. difficile* infection (CDI) to be a major health threat (with 223,900 national cases) among hospitalized patients, eventually leading to 12,800 deaths [1]. In France, in 2016, CDI incidence in acute care was estimated to be 3.6 cases per 10,000 patient days [2]. In 2016, a total of 7711 CDI cases were reported to the ECDC (European Centre for Disease Prevention and Control) in Europe (20 EU countries), of which 74.7% were associated with health care settings [3].

In this review, two situations are distinguished in order to understand the trigger symptoms: CDI and AC of *C. difficile*. CDI is defined by the presence of diarrhea and at least one other following criterion: the carriage of a toxigenic strain of *C. difficile*, the presence of toxins in the stool and/or a colonoscopy result showing pseudomembranous colitis. AC by *C. difficile* can be defined as the absence of diarrhea with at least one other criterion: carrying a strain of *C. difficile* and/or the presence of toxins in the stool [4]. Crobach et al., defined AC as the presence of *C. difficile* but without symptoms of CDI.

The main risks factors are PPI use, antibiotics use, corticoid use, hospital stay and age.

PPI use is a risk factor for rCDI (recurrent CDI) [5]. Their use increased stomach pH from 1.2 to 5, raising the possibility of developing CDI or even carrying the bacterium asymptomatically compared with subjects without this treatment [6]. The increase in stomach pH to a value of 5 during digestion does not influence the resistance of the spores of *C. difficile*, as they are able to survive at a normal stomach pH [6,7,8]. The vegetative cells of *C. difficile* can survive in the gastric content only if the pH is equal to or greater than 5 [7].

The use of antibiotics is a well-known risk factor for *C. difficile* asymptomatic colonization or infection due to modification of the gut microbiota. Most of these cases are associated with the use of four antibiotics: clindamycin, cephalosporins, carbapenem and fluoroquinolone [9,10,11]. Other antibiotics, including macrolides, sulfonamide, trimethoprim and penicillin, are less associated with CDI [9,10,11]. Antibiotic exposure increases the possibility of *C. difficile* colonization by 3.7-fold [12] and of developing CDI by 3.55-fold [9]. A previous study showed that among antibiotic-associated diarrhea (AAD) cases, the incidence of CDI was 1.14–1.89% [13], and the frequency of toxigenic *C. difficile* carriage was 18.1–19.0% [13]. 

The use of corticosteroids increases the risk of *C. difficile* colonization in adults admitted to the hospital [14] and immunosuppressive therapy is a risk factor for complicated CDI [15]. Immunocompromised patients have a higher risk of developing rCDI during hospitalization [16].

A recent hospitalization or a recent intensive care unit stay increases the risk of developing CDI by 2.2 and 6.5, respectively [11]. Hospitalization in the previous 6 months increases the risk of colonization by 2.18 [14]. Previous studies showed that in an ambulatory group (*n* = 43), in patients with short hospital stays (*n* = 48) and in patients with long hospital stays (*n* = 102), the percentages of *C. difficile* carriage were 9.5%, 8% and 13%, respectively [17]. CDC hospitalization rates are significantly higher among those 65 years of age and older (by 4-fold) and those over 85 years of age (by 10-fold) compared to those under 65 years [18,19]. 

The percentage of AC evolves as a function of age. It is high in the first months of life and decreases until adult age, and then it increases with advancing age. The percentage of individuals with AC over time is shown in Figure 1 [4,20,21,22,23,24,25,26,27]. Patients aged >65 years old have a 10-fold higher risk of developing CDI than patients in the other age groups [28,29]. 

There are three lines of defense against pathogens: the epithelial barrier, innate immunity, and adaptive immunity. The first step is intestinal colonization by *C. difficile*. Once the bacterium can produce toxins, these toxins transgress the epithelial barrier through the activation of Rho glycosylation, which causes disruption of tight junctions. Secondly, pathogen-associated molecular patterns (PAMPs) are recognized by pattern recognition receptors (PRRs). This interaction induces a rapid innate immunity response. Finally, adaptive immunity provides a highly specific immune response against *C. difficile* [30]. 

The intestinal mucus is composed of two types of mucins: MUC1 (cell-surface) and MUC2 (secreted forms). Secreted MUC2 is found mostly in the feces of healthy people, while people who suffer from CDI have an imbalanced mucus composition; their stool mucus is composed mainly of MUC1, with significantly decreased MUC2 levels [31]. They also present an increase in terminal galactose residues (a known receptor for *C. difficile* toxin A in mice, hamster, rabbits and pigs) and N-acetyl glucosamine (GlcNAc) [31] and a decrease in N-acetyl galactosamine (GalNAc). 

Essential to *C. difficile* spore germination is the presence of primary bile acids (PBAs). PBAs is produced by the liver, is discharged into the small intestine and helps with fatty digestion [20]. PBAs stimulates germination of *C. difficile* spores [32], and secondary bile acids (SBAs) inhibit germination [32].

Several studies have tried to understand the *C. difficile* pathogenesis in order of reduce the risks of development of the disease and find new therapeutic strategy. The animal experimentations using hamster have allowed to test the transplantation fecal efficiency, the use of non-toxigenic strain of *C. difficile* and use of monoclonal antibodies against toxin A and toxin B [33,34,35]. The piglet model of CDI is representative of the key characteristics of human CDI and helped to understand the virulence and new treatment [36]. Then, *C. difficile* studies use different methods *in vitro* in order to limit using animal experimentation: feces cultures [37], the continuous culture model [38,39], the triple-stage chemostat human gut model [40,41], and the Tim-2 model [42].

The main objective of this review is to explore the available data about the link between the gut microbiota and the development of CDI. The secondary aim is to provide more information on why some people colonized with toxigenic *C. difficile* develop CDI and others show no signs of disease.

## 2. Microbiota Associated with Asymptomatic Colonization and CDI

### 2.1. Composition of the Normal Human Gut Microbiota

The composition of the gut microbiota is influenced by diet, age, the use of antibiotics, etc. [43]. In a normal gut, the dominant phyla are Firmicutes, Bacteroidetes, Actinobacteria, Proteobacteria, Fusobacteria, and Verrucomicrobia. Firmicutes and Bacteroidetes represent 90% of the gut microbiota [44,45]. Gut Bacteroidetes is mainly composed of two genera, *Bacteroides* and *Prevotella* [44,45]. The main genera in the Firmicutes phylum are *Lactobacillus*, *Bacillus*, *Clostridium*, *Enterococcus*, *Faecalibacterium*, *Roseburia*, and *Ruminococcus*. The Actinobacteria phylum is represented mainly by the *Bifidobacterium* genus [44,45]. The Metahit consortium classified human fecal metagenomic samples from three continents into three groups called “enterotypes”: *Bacteroides* (enterotype I), *Prevotella* (enterotype II) and *Ruminococcus* (enterotype III) [45,46]. The enterotype concept is very controversial in the scientific community [46]. Gorvitovskaia et al., chose to validate the first two enterotypes (*Prevotella* and *Bacteroides*) [47], while Cheng et al., have shown that enterotypes are not constant over time and are influenced by age and diet [43].

The gut microbiota of infants is rich in *Bifidobacterium* spp. and the gut microbiota of elderly individuals has decreased proportions of *Ruminococcaceae*, *Bifidobacterium*, *Lactobacillus* and *Faecalibacterium* and increased proportions of Proteobacteria, Bacteroidetes and *Clostridium* spp. [44,48,49].

### 2.2. Factors Influencing the Healthy Gut Microbiota

The use of proton pump inhibitors (PPIs) influences the pH of the stomach and therefore the gut microbiota. The use of PPIs decreased gut microbial diversity and the abundances of *Clostridiales* and *Ruminococcaceae* [50,51] and increased the abundances of the *Enterococcaceae* and *Staphylococcaceae* families [50,52] and *Veillonella parvula* and *Streptococcus mutans* [53]. These modifications of the gut microbiota are strongly correlated with CDI development.

In human feces, clindamycin, cephalosporins and fluoroquinolone treatments impact the microbiota, resulting in an increase in *Enterococcus* abundance [54]. These three categories of antibiotics induce a reduction in the abundances of *Streptococcus* spp., *Anaerococcus* spp., *Peptoniphilus* spp., *Porphyromonas* spp. and *Prevotella* spp., and increase the abundance of *Sphingomonas* spp. [54]. Antibiotics also seem to decrease the proportions of *Ruminococcaceae*, *Lachnospiraceae* and *Bifidobacterium* and to increase the proportions of *Lactobacilliaceae* and *Streptococcaceae* [49]. Werkhoven and collaborators (2021) showed that the carriage of toxigenic *C. difficile* increased the incidence of developing CDI 10-fold after antibiotic treatment. Specifically, the use of carbapenem increased the incidence of CDI 5-fold and increased the abundance of *Enterococcus* 5-fold [13].

The age-modified gut microbiota. The gut microbiota of elderly individuals has decreased proportions of *Ruminococcaceae*, *Bifidobacterium*, *Lactobacillus* and *Faecalibacterium* and increased proportions of Proteobacteria, Bacteroidetes and *Clostridium* spp. [44,48,49]. Two interesting recent studies have addressed the gut microbiota composition in hospitalized elderly patients with CDI, showing lower microbial diversity, lower proportions of gut commensals with putative functions and a reduction in butyrate-producing bacteria in CDI samples [55,56].

The state of dysbiosis can be defined as a decrease in the obligate anaerobic bacteria and an increase in the relative abundance of facultative anaerobic bacteria, such as *Enterobacteriaceae* [57,58], a decrease in microbial diversity and a decrease in anti-inflammatory species such *Faecalibacterium prausnitzii* [59]. In Inflammatory Bowel Disease (IBD), a decrease in microbial diversity, a decrease in *F. prausnitzii* and an increase in *Streptococcus* and *Escherichia*/*Shigella* are observed [59,60].

### 2.3. Composition of Microbiota among Patients with AC

As previously described, AC by *C. difficile* can be defined as the absence of diarrhea with at least one other criterion: carrying a strain of *C. difficile* and/or the presence of toxins in the stool [4]. In the literature, few studies differentiate between AC by *C. difficile* and CDI. The composition of the microbiota of patients with AC was similar to that of the control group and included the phyla Bacteroidetes (40.95%), Firmicutes (36.23%), and Proteobacteria (15.73%) [61].

Within the phylum Bacteroidetes, decreases in two families (*Bacteroidaceae* and *Prevotellaceae*) were observed [17]. Zhang et al., showed decreases in the AC of *Prevotella* spp., *Alistipes* spp., *Bacteroides* spp. and an increase in the AC of *Parabacteroides* spp. [61].

In the phylum Firmicutes, increases in the abundances of *Ruminococcaceae*, *Erysipelotrichaceae* and *Clostridiaceae* and decreases in the abundances of *Leuconostocacceae* and *Erysipelothrichaceae* were observed [17]. Zhang et al., showed decreases in AC by *Dorea* spp., *Coprococcus* spp., and *Roseburia* spp. [61] and increases in AC by *Lactobacillus* spp. [61], *Enterococcus* spp. [51] and *Oscillospira* spp. [62]. Another study showed increases in AC by *Blautia* spp., *Flavonifractor* spp., and *Lachnospiraceae*_unclassified [63].

Within the phylum Proteobacteria, several studies showed an increase in AC by *Enterococcus* spp. and *Klebsiella* spp. [61]. In the phylum Actinobacteria, Zhang et al., showed a decrease in AC by *Bifidobacterium* spp. [61]. Within the phylum Verrucomicrobia, a decrease in AC by *Akkermansia* spp. was observed [63].

Some studies showed no differences in microbial diversity between an AC group and a healthy group (HG) who presented a negative stool test for *C. difficile* [63]; however, some studies showed that the microbial richness (Chao index) decreased in patients with AC compared with patients in the HG [61].

### 2.4. Microbiota Composition of Adults Suffering from CDI

Many studies have described the gut microbiota composition of patients with CDI. In Table 1, the modifications of gut microbiota with increases (red) and decreases (green) in the abundances of various phyla and genera when a patient is suffering from CDI are represented. These data are from the original research publications.

The prevalence of the phylum Proteobacteria is increased in adults with CDI [61,62,64,65,66]. The main family responsible for this increase is *Enterobacteriaceae* [62,66,67,68,69], and the main genera exhibiting increases are *Klebsiella* spp. [56,61,62,66,70], *Escherichia*/*Shigella* [56,61,64,66,69,70,71], *Proteus* spp. [56,66] and *Providencia* spp. [66]. 

The prevalence of the phylum Firmicutes is decreased in adults with CDI [61,62,65]. The main families responsible for this decrease are *Lachnospiraceae* and *Ruminococcaceae* [54,56,62,66,67,72], and the main genera are *Blautia* spp. [54,61,64,66,70,73,74], *Roseburia* spp. [54,61,64,66,74], *Anaerostipes* spp. [62,64,74], *Faecalibacterium* spp. [54,56,61,62,64,66,69,70,74] *Collinsella* spp. [56,64,66,69,70] and *Coprococcus* spp. [61,62,69,70,74]. Some of the genera that have been shown to exhibit increases are *Enterococcus* spp. [54,56,61,62,69,70,71], *Veillonella* spp. [56,61,62,74] and *Lactobacillus* spp. [61,70,71,74]. Metabolization of PBAs to SBAs is provided by populations from the Firmicutes phylum: *Ruminococcaceae*, *Blautia* and *Lachnospiraceae* [20,75]. The decreases in *Lachnospiraceae* and *Ruminococcaceae* also lead to decreases in the concentration of SCFA and the transformation of SBAs from PBAs, providing advantages to these bacteria [20,56,62,64,67]. If the abundances of these bacterial groups decrease, there is a decrease in SBAs and therefore a decrease in the inhibition of germination in the ileum. This decrease in SBAs facilitates the development of CDI. *Clostridium scindens* is able to restore SBAs metabolism and inhibit the germination of *C. difficile* [20,32,76].

Regarding the phylum Bacteroidetes, some studies showed a general decrease in abundance [61,62,67] while others showed an increase [64,68]. Depending on the studies, the abundance of the genus *Bacteroides* spp. has been shown to decrease [54,56,61,71] or increase [62,64,74]. Only one study showed that the abundance of *Parabacteroides* spp. increased [61]. Some genera, such as *Prevotella* spp. [54,61,62,64,70,73], *Paraprevotella* spp. [66], *Alistipes* spp. [54,56,61,69,70], and *Porphyromonas* spp., showed decreased abundances [54].

The abundance of the phylum Actinobacteria seems to decrease with adult CDI [61,62,64]. The main genus responsible for the decrease is *Bifidobacterium* spp.

Within the phylum Verrucomicrobia, some studies showed that an increase in the abundance of *A. muciniphila* is associated with CDI [62,68,71,77], while others reported that an increase in the abundance of *Akkermansia* protected against CDI [56,63,69].

Hernandez et al. (2019) classified CDI into two groups according to prognosis. Cluster A showed high abundances of *Enterococcaceae* and *Enterococcus* and decreases in the abundances of *Bacteroidaceae* and *Lachnospiraceae*. This cluster was associated with more severe diarrhea, more aggressive therapy and a poor prognosis. Cluster B had high abundances of *Bacteroidaceae* and *Lachnospiraceae*. This cluster was associated with less severe diarrhea, less aggressive therapies and a good prognosis [67].

Several recent studies have focused on comparing the gut microbiota of people with CDI with people with negative *C. difficile* detection and symptomatology. A significantly decrease in the bacterial population diversity (Shannon index) [54,64,65,66,70,74] and a significantly lower richness (Chao1 index) [64,66] has often been observed in patients with CDI.

People who developed a single CDI had higher levels of IgM anti-toxin A, toxin B and non-toxigenic antigens on Day 3 and significantly higher IgG anti-toxin A on Day 12 than people who developed recurrent CDI forms [30].

In Table 2, more details about these studies are provided.

**Table 1 pathogens-11-00781-t001:**
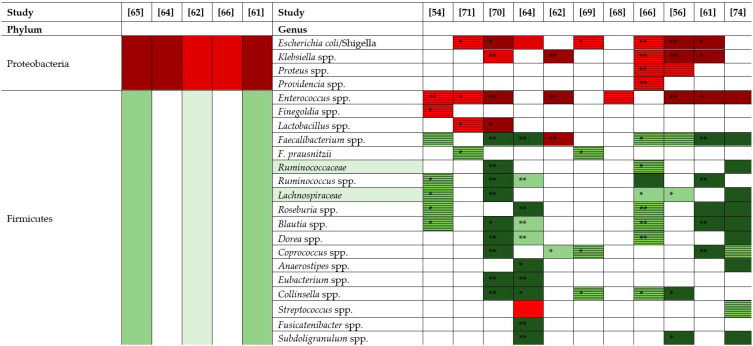
Variation in the abundance of the main phyla and genera of the gut microbiota of CDI people versus healthy people.

*p*-values are the original value from research article, where * *p* < 0.05 ** *p* < 0.001. Bright red (
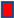
) represents an increase of +10 to 33 %, red (
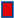
) an increase of +34 to 66%, dark red (
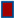
) an increase of +67 to 100%. Bright green (
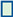
) represents a decrease of −10 to −33%, green (
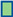
) a decrease of −34 to −66%, dark green (
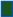
) a decrease of −67 to −100%. Striped red (
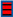
) represents an increase by unspecified value and striped green (
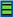
) a decrease by unspecified value.

## 3. Fecal Microbiota Transplantation (FMT)

FMT is the transfer of the fecal microbiota containing bacteria from a healthy volunteer into a diseased patient [82,83]. In 1958, Dr Ben Eiseman described the “fecal enema” as a treatment for pseudomembranous enterocolitis [83]. FMT is administered in several ways: with capsules for ingestion, with an endoscope and with a nasoenteric tube [84]. The criteria for selecting a healthy donor have been described in several papers [84,85]. Briefly, the blood and the stool must be tested to ensure that no infectious diseases or pathogenic bacteria are present, and a series of criteria are checked, including recent antibiotic use, history of diarrhea, and history of immune disorders [84,85].

In the case of primary CDI, this treatment could be used before using antibiotics or in addition of antibiotics to avoid rCDI [86]. In the case of rCDI, FMT is the second line of treatment. FMT has been reported to be successful in 80–92% of patients with rCDI [84,87,88,89] and with primary CDI [89]. This treatment is safe and effective in adults [87,88], in elderly [90,91,92] and in children [93].

Regarding the composition of the gut microbiota of patients with FMT, the alpha diversity (Shannon index) seems to be lowest pretreatment among patients with CDI, and the alpha diversity is restored after FMT [87,88,90,91]. The composition of the microbiota has an impact on the recurrence of CDI and the success of FMT. After FMT, the phylum Firmicutes increased significantly in rCDI (<65 y) and the phylum Proteobacteria decreased significantly in rCDI (>65 y) [90]. *Lachnospiraceae*, *Ruminococcaceae* and *Bifidobacteriaceae* increased significantly [90] and *Enterobacteriaceae* decreased significantly after FMT in rCDI [89,90]. *Blautia*, *Ruminococcus*, *Coprococcus*, *Bifidobacterium* [90] and *Faecalibacterium* [89] spp. increased significantly after FMT in rCDI [89,90]. These modifications of the gut microbiota after FMT strongly suggested that these bacterial populations are associated with healthy people (See Table 1) and will favor a good prognosis. Staley et al. (2018) showed that a follow-up analysis of 16S rDNA extracted from feces can be used to predict an eventual recurrence of CDI. After FMT, high proportions of *Lactobacillales*, *Enterobacteriaceae*, *Enterococcus*, *Klebsiella*, *Streptococcus* and *Veillonella* and reductions in *Roseburia*, *Blautia*, *Lachnospiraceae*, *Ruminococcaceae*, *Anaerostipes*, *Coprococcus*, *Dorea* and *Clostridiales* [88,91] will disadvantage the patient and promote rCDI. These bacterial population are associated with the gut microbiota of CDI cases (see in Table 1). 

Before using the FMT, a preventive probiotic administration before the antibiotics use is effective [94]. The use of probiotics before and at the same time as antibiotics reduces the risk of CDI by >50% in hospitalized adults [94].

## 4. Conclusions

Many studies have characterized the gut microbiota composition of patients with CDI, but confusion is still present in the literature between CDI and AC. Few studies have differentiated AC by *C. difficile* and CDI. This review explores the available data related to the link between the gut microbiota and the development of *C. difficile* infection. The causes of the development of CDI are clearly multifactorial. An external cause (such as a medication) can disrupt the homoeostasis of the gut microbiota. PPI and antibiotic use decrease the richness of the gut microbiota [95]. This imbalance promotes the growth of some bacteria (for example, *A. muciniphila*), and these bacteria can degrade the mucus layer and allow the pathogenic bacteria and toxins access to the epithelium. Moreover, the abundances of some bacteria (*Lachnospiraceae*, *Ruminococcaceae* and *Blautia*) decrease, and these bacteria are involved in bile metabolism and can increase the primary bile acid concentration. Higher PBAs concentrations are favorable to *C. difficile* germination and multiplication. Some bacteria also have a positive correlation (*Enterococcus*, *Enterobacteriaceae*) or negative correlation (*Blautia*, *Prevotella*, *Roseburia*, *Dorea*, *Collinsella*, *Coprococcus*, *Ruminococcus*, *Ruminococcaceae*, *Lachnospiraceae*) with *C. difficile* colonization and/or CDI. The gut microbiota will promote the development of the CDI. Through all the studies, the CDI has a gut microbiota footprint with the decrease and the increase in some bacteria. In this review, a lot of bacteria are singled out for giving an advantage or a disadvantage when developing CDI. Some of these bacteria have an impact on gut health. *Faecalibacterium prausnitzii* is considered as a species of the healthy gut microbiota. This bacterium is reduced in gut dysbiosis, in IBD, in obesity, in diabetes, etc. [96]. *Lachnospiraceae* is protective against CDI [20]. *C. scindens*, member of *Lachnospiraceae* have a protective effect against CDI [20,32]. Amrane et al. (2018) showed that *C. scindens* is present in the feces when patient developing CDI [97]. *Alistipes* spp. indicated a controversial pathogenicity. On the one hand, the bacteria have protective effects against liver fibrosis, cancer immunotherapy and cardiovascular disease [98]. On the other hand, this genus is associated with colorectal cancer and mental disease [98].

In this review, CDI can be associated with an increase or a decrease in *A. municiphila* and AC is associated with a decrease *A. municiphila*. The presence of this bacteria is positive against obesity, diabetes, cardiometabolic disease and low-grade inflammation [99]. It is actually used to manage obesity [100]. In human intestinal organoids, *C. difficile* is capable of decreasing MUC2 production, but it is not responsible for altering host mucus oligosaccharide composition [31]. Furthermore, it has been reported that *C. difficile* is not capable of degrading mucin glycans, although coculture with mucin-degrading *Akkermansia muciniphila*, *Bacteroides thetaiotaomicron* and *Ruminococcus torques* allowed the pathogen to grow in media that lacked glucose but contained purified MUC2 [101]. When mucus is degraded by bacteria, oligosaccharides (GlcNAc, GalNAc, galactose, mannose and fucose) are salted out [101], and *C. difficile* is capable of using these oligosaccharides [31].

The *Enterobacteriaceae* family is associated with the dysbiosis state [58,59]. *Enterococcus* spp. is a controversial bacterium [102]. It is a commensal bacterium of intestinal flora, vagina, and mouth microbiota [102]. *E. faecium* and *E. faecalis* are used as probiotics [102,103] and *Enterococcus* spp. is used in meat and cheese [102,104] fermentation. Recently, it was shown that *E. faecalis* and *E. faecium* are potentially pathogenic bacteria due to their ability to adapt in new environment [102,105]. Additionally, a resistance to vancomycin has emerged in this genus [102,105]. Romyasamit et al. (2020) exhibited that six *E. faecalis* strains have a probiotic effect and anti-*C. difficile* activity [106]. *Klebsiella pneumonae* is present in the mucus layer with *C. difficile* [107].

The second objective of this review was to provide more information on why some people colonized with toxigenic *C. difficile* develop *C. difficile* infection and others show no signs of disease. The answer to this question is still unknown, but some facts will improve the understanding. The response of the adaptative immune system impacts the development of the disease. Patients exhibiting AC were shown to have higher antibody levels against *C. difficile* than people who developed CDI. It has been reported that sixty percent of the general population has had AC or has been infected with *C. difficile*, as determined by the observation of detectable seric IgG and IgA antibodies to toxins A and B [108]. IgG and IgA titers against toxins A and B are significantly higher in children positive for toxigenic strains than in people carrying non-toxigenic strains [109]. IgG antibody levels against toxin A are significantly higher within 3 days of colonization in AC patients than in those who develop CDI [30,108,110]. IgG levels against toxin B and non-toxin antigens seem to be higher among individuals who develop AC [30,108,110]. 

Some risks factors will predispose patients to developing CDI (antibiotics use, PPI use, age, etc.). The decrease in the diversity described in elderly gut microbiota [18], the decrease in some bacterial population (*Ruminococcaceae*, *Bifidobacterium*, *Faecalibacterium*) and the increase in some bacterial population (Proteobacteria, Bacteroidetes and *Clostridium* spp.) suggest why this population have a 10-fold higher risk of developing CDI. 

Some treatments involving bacteria are commonly used and effective against CDI. FMT allows the recovery of patients with recurrent CDI with an increase in the abundances of some bacteria (*Blautia*, *Collinsella*, *Anaerostipes*, *Coprococcus*, *Dorea* and *Roseburia*) and a decrease in the abundances of others (*Lactobacillales*, *Enterobacteriaceae*; *Enterococcus*, *Klebsiella*, *Streptococcus* and *Veillonella*). More research with strict inclusion criteria is needed to measure AC and CDI gut microbiota analysis. The purpose of this work was to study the impact of in vivo control measures for the gut microbiota to decrease colonization and CDI or to improve recovery. 

## Figures and Tables

**Figure 1 pathogens-11-00781-f001:**
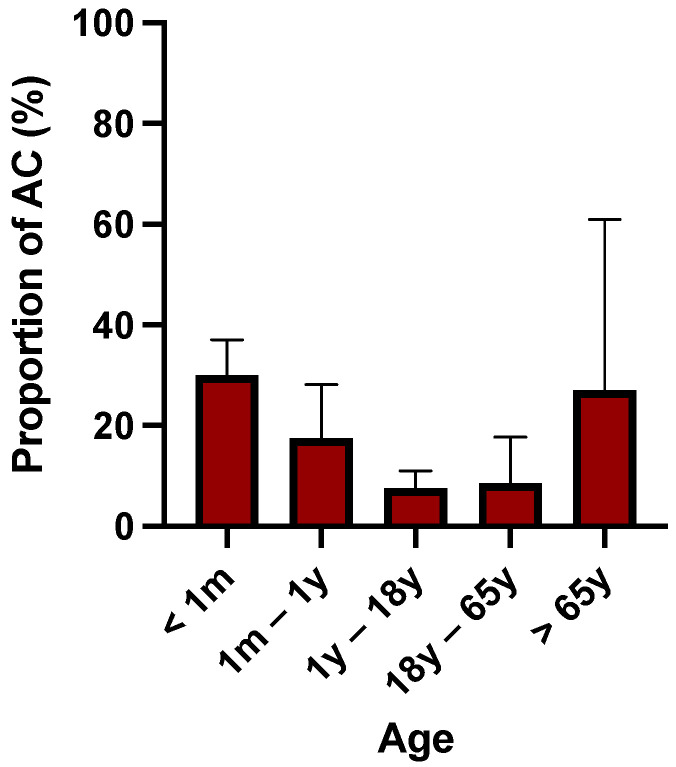
Box plot illustrating the mean relative proportion of *C. difficile* asymptomatic carriers in function of category of age. These data were found in articles that studied the prevalence of AC [4,20,21,22,23,24,25,26,27].

**Table 2 pathogens-11-00781-t002:** Analysis of the most important taxa exhibiting increased and decreased abundances in the CDI group compared with the control group.

Analysis Method	Study	Study Group	Diversity	Increase in CDI Group Compared with Control Group	Decrease in CDI Group Compared with Control Group
16S rRNA gene amplicon analysis ^a^ Illumina	[54]	CDI group (*n* = 15) Control group (*n* = 669) Mean age: 70 y Location: Europa	The alpha diversity (Chao1 and Shannon index) was lower in the CDI group. ^*^	*Enterococcus* spp. *	*Bifidobacterium* spp., *Blautia* spp. *Faecalibacterium* spp., *Bacteroides* spp. and *Prevotella* spp.
qPCR	[71]	CDI group (*n* = 28; 79 y) Control group (*n* = 56; 75 y) Country: Iran	NR	*A. muciniphila* *, *Lactobacillus* spp. * and *Escherichia coli* *	*Bacteroides* spp. *, *Bifidobacterium* spp.* and *F. prausnitzii* *
16S rRNA gene amplicon analysis ^a^ Illumina HiSeq	[70]	CDI group (*n* = 26; 66.5 y) Control group (*n* = 61)	The alpha diversity (Shannon index) was lower in the CDI group. *	*Enterococcus* *, *Lactobacillus* *, *Escherichia* * and *Klebsiella* *.	*Bifidobacterium* *, *Ruminococcus* *, *Eubacterium* *, *Faecalibacterium* spp.*, *Prevotella* *, *Blautia* *, *Collinsella* *, *Dorea* *, *Alistipes* *, *Lachnospiraceae* * and *Coprococcus* * (*p* < 0.05)
16S rRNA gene amplicon analysis ^a^ MiSeq Illumina	[65]	CDI group (*n* = 11; 70.81 ± 20.1 y) Control group (*n* = 8; 18 to 45 y) Country: France	The alpha diversity (Shannon index) was lower in the CDI group. *	*Proteobacteria* *^ns^	*Firmicute* *^ns^ *Actinobacteria* *^ns^ *Bacteroidetes* *
454 pyrosequencing analysis of bacteria ^e^ 454 GS FLX Titanium Sequencing System (Roche)	[64]	CDI group (*n* = 24, 64.8 ± 15.7 y) Control group (*n* = 13, 49.2 ± 11.5 y). Country: Korea	The richness (Chao1 index; 283.3 vs. 642.9) and the alpha diversity (Shannon index; 3.6 vs. 4.5) were lower in the CDI group. *	*Proteobacteria* *^ns^ *Bacteroidetes* *^ns^ *Verrucomicrobia* *Bacteroides*, *Streptococcus* spp., and *Escherichia*	*Actinobacteria* * *Blautia*, *Bifidobacterium*, *Faecalibacterium*, *Collinsela*, *Dorea*, *Eubacterium*, *Fusicatenibacter*, *Prevotella*, *Roseburia*, *Subdoligranum*, *Ruminococcus*, *Clostridium*, *Catenibacterium*, Dialister *and Anaerostipes*
16S rRNA gene amplicon analysis Metatranscriptomic Illumina Hiseq4000 platform	[69]	CDI group (*n* = 18, 65,3 ± 17 y) Control group (*n* = 31, 60 ± 18 y) Country: United States	A lower average species evenness was observed in the CDI group with Heip’s evenness (93.4 ± 23.1 in the CDI group vs. 121.8 ± 58.2 in the control group). *^ns^	Metagenetic Increases in the proportions of *Clostridiaceae*, *Peptostreptoccocaceae*, and *Enterococcus* Metatranscriptomic Increases in the proportions of *Clostridioides difficile*, *E. coli*, unclassified *Peptostreptococcaceae*, and *Enterobacteriaceae*	Metagenetic Decreases in the proportions of *Faecalibacterium* and *Collinsella* Metatranscriptomic Decreases in the proportions of *A. municiniphila*, *Faecalibacterium prausnitzii*, *Coprococcus*, *Alistipes shahii*, *Collinsella and Verrucomicrobiaceae* *
16S rRNA gene amplicon analysis ^b^ MiSeq technology Illumina	[68]	CDI group (*n* = 13; 55.5 ± 20.5 y) Control group (*n* = 13; 51.2 ± 16.6 y)	No difference was observed in richness or evenness between groups. *^ns^	*Enterobaceriaceae* * *Peptostreptococcaceae* * *A. muciniphila* *	*Bacteroidales* *Clostridales*
16S rRNA gene amplicon analysis ^c^ Ion Chez System and Ion S5 L system	[62]	*tcdB*-positive group (*n* = 79, 62.5 ± 19.9 y) divided into two groups: CDI group (*n* = 58) and colonized group (*n* = 21). Control group (*n* = 20, 62.2 ± 14.4 y). Country: Korea	The richness (Chao1 index; 60 vs. 95) was lower in the *tcdB* group. *	*Proteobacteria* *, *Enterobacteriaceae**, *Porphyromonadaceae* *, *Enterococcaceae* * *Parabacteroides**, *Enterococcus* *, *Veillonella* *, *Klebsiella* * and *Akkermansia* *	*Lachnospiraceae* *, *Ruminococcaceae* *, *Prevotellaceae* * *Prevotella*, *Phascolarctobacterium*, *Haemophilus*, *Lachnospira*, *Coprococcus*, *Dialister*, *Butyricimonas*, *Catenibacterium*, *Faecalibacterium*, *Paraprevotella*, *Odoribacter and Anaerostipes*
Pyrosequencing ^e^ Roche GS Junior	[63]	Control group (*n* = 94) CA group (*n* = 24) Mean age: 78.66 y	No difference in diversity between the groups.	*Blautia* *Flavonifractor* *Lachnospiraceae*_unclassified	*Akkermansia*
Pyrosequencing ^d^ Illumina MiSeq	[77]	CA group (*n* = 7) Control group (*n* = 25) Mean age: 89.3 y Country: United States.	No difference in the alpha diversity or beta diversity (Shannon index: 3.47 in the CA group vs. 3.12 in the control group).	*Firmicutes*, *Actinobacteria* *Akkermansia* spp., *Dermabacter* spp., *Romboutsia* spp., *Meiothermus* spp., *Peptoclostridium* spp., *Ruminococcaceae* UGC 009 *	*Bacteroidetes*, *Proteobacteria*
Pyrosequencing ^a^ Roche 454 GS FLX and sequencer	[61]	CDI group (*n* = 8, 58.9 ± 22.2 y) CA group (*n* = 8, 60.5 ± 20.8 y) Control group (*n* = 9, 60.8 ± 16.02) Country: China	Reductions in richness (Chao) *^ns^ and diversity (Simpson and Shannon indexes) * in the CDI and CD+ groups.	*Proteobacteria* *Fusobacteria* *Clostridium* cluster XI *Parabacteroides* *Escherichia/Shigella* *Klebsiella* *Enterococcus* *Veillonella* *Lactobacillus*	*Bacteroidetess and Firmicutes.* *Prevotella*, *Bacteroides*, *Faecalibacterium*, *Coprococcus*, *Roseburia*
Pyrosequencing ^d^	[17]	Control group (*n* = 252) CA group (*n* = 22) Age: 65 y Country: Ireland	NR	*Bacteroidaceae* *Ruminococcaceae* Clostridiaceceae *Erysipelothrichaceae* *Aerococacceae* (one patient) *Flavobacteriaceae* (one patient)	*Enterococcaceae* *Prevotellaceae* *Leuconostocacceae* *Spirochaetaceae*
16S rRNA gene amplicon analysis ^b^ Illumina MiSeq sequencer	[56]	CDI group (*n* = 25, 82.9 ± 8.5) AB group (*n* = 29, 84.2 ± 8.1) Control group (*n* = 30, 82.3 ± 6.8). Country: Italy	Decreased diversity index (Chao1 and Shannon index) in the CDI group compared to the AB group.	*Klebsiella* *, *Escherichia/shigella* *, *Sutterella*, *Enterococcus* *, *Citrobacter*, *Veillonella*, *Proteus*, *Morganella*, *Hafnia*, *Corynebacterium*, *Staphylococcus*	*Faecalibacterium*, *Bifidobacterium*, *Akkermansia*, *Bacteroides* *, *Lachnospira* *, *Alistipes* *
16S rRNA gene amplicon analysis ^b^ QMiSeq Illumina	[66]	CDI group (*n* = 15; 61 y). Diarrhea group (*n* = 18; 56.5 y). Control group (*n* = 25, 58 y) Country: China	Decreased alpha diversity (Shannon index) and richness (Chao1) in the CDI group and diarrhea group. *	*Proteobacteria* *, *Enterococcaceae*, *Streptococcaceae*, *Lactobacillaceae* and *Peptostreptococcaceae* *^ns^ *Clostridium* spp.*^ns^ *Actimnomyces* and *Rothiabacterium* *Escherichia/Shigella*, *Klebsiella*, *Proteus* and *Providencia*	*Firmicutes and Bacteroidetes* * *Lachnospiraceae*.* *Blautia*–*Lachnospiraceae*_incertae_sedis, *Roseburia* and *Dorea* *Ruminococcaceae*.* *Faecalibacterium*, *Clostridium* IV, *Oscillospira* and *Ruminococcus*
454-pyrosequencing ^f^ FLX Titanium platform	[73]	All samples with diarrhea CDI–toxins group (71 ± 15 y) CDI–without toxins group (66.3 ± 27 y) Control group (52.0 ± 13 y) Country: Spain	NR	*Bacteroides* *Parabacteroides* *Faecalibacterium* *Clostridium* XIVa *Clostridium* cluster XI CDI group with and without toxins: *Phascolarctobacterium*, *Enterococcus*, *Clostridium* XI cluster, *flavonifractor*, and *Erysipelotrichaceae incertea sedis* * CDI group with toxins: *Enterococcus*, *Clostridium XI* cluster, and *Erysipelotrichaceae incerteae sedis*	CDI group with and without toxins: *Blautia*, *Holdemania*, *Enterobacteriaceae*, and *Veillonellaceae* * CDI group with toxins: *Fusobacterium*, *Prevotella*, *Vovibrio*, and *Dialister*
454-Pyrosequencing ^g^	[78]	CDI group (*n* = 94; 55.9 ± 18.3) CDN (=*C. difficile-negative* nosocomial diarrhea) group (*n* = 89; 58.7 ± 14.9) Control group (*n* = 155; 52.2 ± 21.5)	Reduction in the diversity (inverse Simpson index) * in the CDI and CDN groups compared with the control group.	*Enterococcus* * *Lachnospiraceae* * *Erysipelotrichaceae* *	*Bacteroides* species *
Pyrosequencing ^e^	[74]	CDI group (*n* = 39; 54.7 ± 20.1) CDN (= *C. difficile-negative* nosocomial diarrhea) group (*n* = 36; 54.6 ± 20.0) Control group (*n* = 40; 60.9 ± 9.1)	Reduction in the diversity (Shannon index) * in the CDI and CDN groups compared with the control group.	*Bacteroides* *Veillonella* *Enterococcus* *Lactobacillus* *Fusobacterium*	Firmicutes *Lachnospiraceae* and *Ruminococcaceae* *Blautia* * *Pseudobutyrivibrio*, *Roseburia*, *Faecalibacterium*, *Anaerostipes*, *Subdoligranulum*, *Ruminococcus*, *Streptococcus*, *Dorea* * and *Coprococcus.*
16S rRNA gene amplicon analysis ^a^ MiSeq technology Illumina	[79]	CDI group (*n* = 57; 69.5 y): GDH+ and TcdB No control group Country: Spain	The richness (Chao1) was 134.32 and the alpha-diversity (Shannon index) was 4.01.	*Bacteroides* (46.51%), Firmicutes (34.70%), Proteobacteria (13.49%). *Bacteroidaceae* (31.01%). *Enterobacteriaceae* (9.82%), *Lachnospiraceae* (9.33%), Tannerllaceae (6.16%) and *Ruminococcaceae* (5.64%)	
16S rRNA gene amplicon analysis ^d^ MiSeq technology Illumina	[80]	CDI group (*n* = 31; 64.0 y). Three periods: pretreatment ATB, two days after treatment, seven days after treatment or discharge. No control group	The alpha diversity (Shannon index) was lower with CDI pretreatment in the recurrent group. *	*Veillonella dispar* * predictor of recurrence.	/
16S rRNA gene amplicon analysis ^d^ Illumina MiSeq sequencer	[81]	CDI group (*n* = 88; 52.7 y) G1 (ATB responder) G2 (non-ATB responder) G3 (recurrent CDI) G4 (non-recurrent CDI) No control group (other studies)	Decreased alpha diversity (Chao1 index) in the CDI group than in the control group.	G1 *Ruminococcaceae; Rikenellaceae; Clostridiaceae; Bacteroides; Faecalibacterium; Rothia* G2 *Clostridiaceae*, *Lachnospiraceae*, *Blautia*, *Coprococcus*, *Streptococcus*, *Bifidobacterium*, *Ruminococcus and Actinomyces.* G3 with recurrent CDI *Veillonella*, *Enterobacteriaceae*, *Streptococci*, *Parabacteroides*, and *Lachnospiraceae*	/

^a^ V3–V4 regions of the 16S rRNA gene. ^b^ V3 region of the 16S rRNA gene. ^c^ V2–V3 regions of the 16S rRNA gene. ^d^ V4 regions of the 16S rRNA gene. ^e^ V1–V3 regions of the 16S rRNA gene. ^f^ V1–V2 regions of the 16S rRNA gene. ^g^ V3–V5 regions of the 16S rRNA gene. * Significant. *^ns^, non-significant.

## Data Availability

Not applicable.

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
