# Peer review of "Gut Microbiota Composition Associated with Clostridioides difficile Colonization and Infection"

_pathogens, 2022, doi:10.3390/pathogens11070781_

Round 1

Reviewer 1 Report

Excellent review article on the topic of CDI and gut microbiota. I would only suggest to describe in more details alteration of microbiota in elderly and how it might affect the risk for CDI development. Suggested literature : https://www.ncbi.nlm.nih.gov/pmc/articles/PMC6354172/

Author Response

Point 1: Excellent review article on the topic of CDI and gut microbiota. I would only suggest to describe in more details alteration of microbiota in elderly and how it might affect the risk for CDI development. Suggested literature : https://www.ncbi.nlm.nih.gov/pmc/articles/PMC6354172/

Response 1: Thank you for your comments

*I added in the introduction: “CDC hospitalization rates are significantly higher among those 65 years of age and older (by 4 times) and those over 85 years of age (by 10 times) compared to those under 65 years (18,19).”

*I also added in 2.2. Factors influencing the healthy gut microbiota: two recent studies about the gut microbiota composition in elderly patients with CDI. “Two interesting recent studies have addressed the gut microbiota composition in hospitalized elderly patients with CDI, showing lower microbial diversity, lower proportions of gut commensals with putative functions and a reduction of butyrate-production bacteria in CDI samples”

  1. Vakili, B.; Fateh, A.; Asadzadeh Aghdaei, H.; Sotoodehnejadnematalahi, F.; Siadat, S.D. Intestinal microbiota in elderly inpatients with Clostridioides difficile infection. IDR 2020, 13, 2723–2731, doi:10.2147/IDR.S262019.
  2. Milani, C.; Ticinesi, A.; Gerritsen, J.; Nouvenne, A.; Lugli, G.A.; Mancabelli, L.; Turroni, F.; Duranti, S.; Mangifesta, M.; Viappiani, A.; et al. Gut Microbiota composition and Clostridium difficile infection in hospitalized elderly individuals: a metagenomic study. Sci Rep 2016, 6, 25945, doi:10.1038/srep25945.

*I added in the conclusion: “The decrease of the diversity described in elderly gut microbiota (18), the decrease of some bacterial population (Ruminococcaceae, Bifidobacterium, Faecalibacterium) and the increase of some bacterial population (Proteobacteria, Bacteroidetes and Clostridium spp.) suggest why this population have a 10 times higher risk of developing CDI.”

Reviewer 2 Report

Clostridioides difficile is a leading nosocomial agent that presents a major threat among healthcare regimens. Varied intensities of pathologies in different patients make it even harder to draw a more evident correlation with the factors responsible for such disparity. However, the crucial association of gut microbiome with Clostridioides difficile infection (CDI) is very widely studied. In the current review the authors have discussed this association in context of positive colonization but irrespective of infection. The comparison of gut microbiota among normal humans, individuals with asymptomatic colonization and CDI is definitely of interest.

Major comments:

1.     The review is mainly divided into two parts; one related to risk factors associated with colonization or CDI and the other part focuses on the comparison of gut microbiota composition among healthy, asymptomatic colonized and infected/affected individuals. The second part seems to be more evolved than the first part of the review. It would be scientifically more engaging if the two important parts were connected appropriately to maintain a proper flow, context and correlation between them. Currently they read as two different disjoint parts of the review. Moreover, the first part doesn’t seem to do much justice with the title and objectives of the review. 

2.     Part 3.4 about FMT has very limited information. This part could have been more elaborate to fit with the review theme of emphasizing the important role of gut microbiota in CDI recurrence. Rather I feel it would be wiser to make a separate part highlighting the use of this gut microbiota knowledge to cure CDI through FMT. To this theme, it would also be worth mentioning some literature on use of probiotics as a developing approach to treat CDI. Such a separate part would be a good conclusion to this review.

Author Response

Point 1: The review is mainly divided into two parts; one related to risk factors associated with colonization or CDI and the other part focuses on the comparison of gut microbiota composition among healthy, asymptomatic colonized and infected/affected individuals. The second part seems to be more evolved than the first part of the review. It would be scientifically more engaging if the two important parts were connected appropriately to maintain a proper flow, context and correlation between them. Currently they read as two different disjoint parts of the review. Moreover, the first part doesn’t seem to do much justice with the title and objectives of the review. 

Response 1: Thank you for your comment. The first part of the review is to put a context of the CDI and the asymptomatic colonization and to find some explanation to the second objective. But the real work of this review is about the microbiota. I reduced and merged this part with the introduction to avoid confusion. I added a section in the microbiota section to talk about the “2.2. factors influencing the gut microbiota”.

List of modifications:

I reduced the paragraph on PPI lines 45-53:The main risks factors are PPI use, antibiotics use, corticoid use, hospital stay and age. PPI use is a risk factor for rCDI (recurrent CDI) [6]. Their use increased stomach pH from 1.2 to 5, raising the possibility of developing CDI or even carrying the bacterium asymptomatically compared with subjects without this treatment [5]. The increase in stomach pH to a value of 5 during digestion does not influence the resistance of the spores of C. difficile, as they are able to survive at a normal stomach pH [5,7,8]. The vegetative cells of C. difficile can survive in the gastric content only if the pH is equal to or greater than 5 [7].”

I deleted the paragraph on PPI: A previous study using a mouse model showed a significant modification of the pH values of the ileum (from 8.07 to 7.6), cecum (from 5.67 to 5.54) and colon (from 5.87 to 5.70) after using PPIs [9].

I reduced the paragraph on ATB lines 54-62:The use of antibiotics is a well-known risk factor for C. difficile asymptomatic colonization or infection due to modification of the gut microbiota. Most of these cases are associated with the use of four antibiotics: clindamycin, cephalosporins, carbapenem and fluoroquinolone [9–11]. Other antibiotics, including macrolides, sulfonamide, trimethoprim and penicillin, are less associated with CDI [9–11]. Antibiotic exposure increases the possibility of C. difficile colonization by 3.7-fold [12] and of developing CDI by 3.55-fold [9]. A previous study showed that among antibiotic-associated diarrhea (AAD) cases, the incidence of CDI was 1.14-1.89 % [13], and the frequency of toxigenic C. difficile carriage was 18.1-19.0 % [13]»

I deleted the paragraph on ATB: In mouse models, after using antibiotics for one week, the average pH values of the ileum, cecum and colon were significantly increased from 8.07, 5.67, and 5.87 to 8.34, 7.47 and 7.94, respectively [9].

Lines 63-66:The use of corticosteroids increases the risk of C. difficile colonization in adults admitted to the hospital [14]and immunosuppressive therapy is a risk factor for complicated CDI [15]. Immunocompromised patients have a higher risk of developing recurrent CDI during hospitalization [16]. »

Lines 67-74:A recent hospitalization or a recent intensive care unit stay increases the risk of developing CDI by 2.2 and 6.5, respectively [11]. Hospitalization in the previous 6 months increases the risk of colonization by 2.18 [14]. Previous studies showed that in an ambulatory group (n=43), in patients with short hospital stays (n=48) and in patients with long hospital stays (n=102), the percentages of C. difficile carriage were 9.5 %, 8 % and 13 %, respectively [17]. CDC hospitalization rates are significantly higher among those 65 years of age and older (by 4 times) and those over 85 years of age (by 10 times) compared to those under 65 years [18,19].”

Lines 75-78:The percentage of AC evolves as a function of age. It is high in the first months of life and decreases until adult age, and then it increases with advancing age. The percentage of individuals with AC over time is shown in Figure 1 [4,26–33]. Patients aged > 65 years old have a 10 times higher risk of developing CDI than patients in the other age groups [34,35].”

I added a section: 2.2. Factor Factors influencing the healthy gut microbiota (Lines 135-165):

The use of proton pump inhibitors (PPIs) influences the pH of the stomach and therefore the gut microbiota. The use of PPIs decreased gut microbial diversity and the abundances of Clostridiales and Ruminococcaceae [50,51] and increased the abundances of the Enterococcaceae and Staphylococcaceae families [50,52] and Veillonella parvula and Streptococcus mutans [53]. These modifications of the gut microbiota are strongly correlated with CDI development.

In human feces, clindamycin, cephalosporins and fluoroquinolone treatments impact the microbiota, resulting in an increase in Enterococcus abundance [54]. These three categories of antibiotics induce a reduction in the abundances of Streptococcus spp., Anaerococcus, Peptoniphilus spp., Porphyromonas spp. and Prevotella spp., and increase the abundance of Sphingomonas spp. [54]. Antibiotics also seem to decrease the proportions of Ruminococcaceae, Lachnospiraceae and Bifidobacterium and to increase the proportions of Lactobacilliaceae and Streptococcaceae [49]. Werkhoven and collaborators (2021) showed that the carriage of toxigenic C. difficile increased the incidence of developing CDI ten-fold after antibiotic treatment. Specifically, the use of carbapenem increased the incidence of CDI five-fold and increased the abundance of Enterococcus 5-fold [13].

The age modified the gut microbiota. The gut microbiota of elderly individuals has decreased proportions of Ruminococcaceae, Bifidobacterium, Lactobacillus and Faecalibacterium and increased proportions of Proteobacteria, Bacteroidetes and Clostridium spp. [44,48,49]. Two interesting recent studies have addressed the gut microbiota composition in hospitalized elderly patients with CDI, showing lower microbial diversity, lower proportions of gut commensals with putative functions and a reduction of butyrate-production bacteria in CDI samples [55,56].

The state of dysbiosis can be defined as a decrease in the obligate anaerobic bacteria and an increase in the relative abundance of facultative anaerobic bacteria, such as Enterobacteriaceae [57,58], a decrease in microbial diversity and a decrease in anti-inflammatory species such Faecalibacterium prausnitzii [59]. In other state of inflammatory like Inflammatory Bowel Disease (IBD), a decrease of microbial diversity, a decrease in F. prausnitzii and an increase of Streptococcus and Escherichia/Shigella are observed [59,60].”

Point 2: Part 3.4 about FMT has very limited information. This part could have been more elaborate to fit with the review theme of emphasizing the important role of gut microbiota in CDI recurrence. Rather I feel it would be wiser to make a separate part highlighting the use of this gut microbiota knowledge to cure CDI through FMT. To this theme, it would also be worth mentioning some literature on use of probiotics as a developing approach to treat CDI. Such a separate part would be a good conclusion to this review.

Response 2: Thank you for your comment. I created another section and added some literature.

Lines 258: I added:In the case of primary CDI, this treatment could be used before using antibiotics or in addition of antibiotics to avoid rCDI [88]. In the case of recurrent CDI (rCDI), FMT is the second line of treatment. FMT has been reported to be successful in 80-92 % of patients with rCDI [86,89–91] and with primary CDI [91]. This treatment is safe and effective in adults [89,90], in elderly [92–94] and in children [95]. »

Lines 265: I added a bibliography 95.

Lines 265-274: I added:  “The composition of the microbiota has an impact on the recurrence of CDI and the success of FMT. After FMT, the phylum Firmicutes increased significantly in rCDI (< 65y) and the phylum Proteobacteria decreased significantly in rCDI (> 65y) [92]. Lachnospiraceae, Ruminococcaceae and Bifidobacteriaceae increased significantly [92]and Enterobacteriaceae decreased significantly after FMT in rCDI [91,92]. Blautia, Ruminococcus, Coprococcus, Bifidobacterium [92] and Faecalibacterium[91] spp. increased significantly after FMT in rCDI [91,92]. These modifications of the gut microbiota after FMT strongly suggested that these bacterial populations are associated with healthy people (See table 1) and will favour a good prognosis. »

Lines 281-283: I added : “Before using the FMT, a preventive probiotic administration before the antibiotics use is effective [96]. The use of probiotic before and at the same time as antibiotics reduces the risk of CDI by >50% in hospitalized adults [96]”.

Reviewer 3 Report

It is a review article focused on “microbiota influence the development of Clostridioides difficile infection”. The topic is of clinical significance. There are some suggestions:

1.      In Abstract, line 13-14 and In Introduction part, line 39 “Two categories can be distinguished: CDI and AC of C. difficile”. Asymtomatic carriage of C. difficile is not another category to CDI. AC and CDI are more likely the same pathophysiology with different disease severity or pattern.

2.      Line 63 “The use of PPIs decreased gut microbial diversity……..” So what is the correlation between these microbiota change with the development of CDI or AC?

3.      The same for line 79-83 “These three categories of antibiotics induce a reduction in the abundances……and Streptococcaceae” what is the correlation between these microbiota change with the development of CDI or AC?

4.      For Fig 1 and line 96-97. “Patients aged > 65 years….. other age groups” Suggest to have more discussion for Fig 1. For example why aged patients had higher rate or AC? Related to microbiota change?

5.      Line 111 “….TcdA and TcdB” If it is referred to protein, suggest use “toxin A and B”. If it is referred to gene, suggest use italic “tcdA, tcdB”

6.      For 2.2. Immunity; line 101-118. What is the correlation between this paragraph with the topic “the microbiota change”?

7.      For the 3. Microbiota associated with asymptomatic colonization and CDI, line 149-239. These are the most important paragraph related to the topic. However the authors just list the findings from previous articles. As a review article suggest to provide more conclusions or suggestions for readers from these articles.

8.      The same for Table 1 and Table 2. Suggest to provide more conclusions or suggestions for readers from these tables.

9.      For the conclusion part, line 271-296. The authors just list again the findings from previous articles. The same for the conclusion part, suggest to provide more conclusions or suggestions for readers from these tables.

Author Response

Point 1: In Abstract, line 13-14 and In Introduction part, line 39 “Two categories can be distinguished: CDI and AC of C. difficile”. Asymtomatic carriage of C. difficile is not another category to CDI. AC and CDI are more likely the same pathophysiology with different disease severity or pattern.

Response 1: Thank you for your comment. I’m agree with you that CDI and AC are the same pathophysiology. But in the literature, there is a lot of confusion between CDI and AC. In this review, we try to understand why the AC will become a CDI or why some people will be directly symptomatic. That why we would like to be clearer in the two definitions of the CDI and AC.

Lines 13-14: I modified: “In this review, two situations are distinguished: C. difficile infection (CDI) and asymptomatic colonization (AC). »

Lines 38-39: I modified: “In this review, two situations are distinguished in order to understand the trigger symptoms: CDI and AC of C. difficile.”

Point 2: Line 63 “The use of PPIs decreased gut microbial diversity……..” So what is the correlation between these microbiota change with the development of CDI or AC?

Response 2: I added a section in the microbiota section to talk about the “2.2 factors influencing the gut microbiota”

Point 3: The same for line 79-83 “These three categories of antibiotics induce a reduction in the abundances……and Streptococcaceae” what is the correlation between these microbiota change with the development of CDI or AC?

Response 3: I added a section in the microbiota section to talk about the “2.2 factors influencing the gut microbiota”

Point 4: For Fig 1 and line 96-97. “Patients aged > 65 years….. other age groups” Suggest to have more discussion for Fig 1. For example why aged patients had higher rate or AC? Related to microbiota change?

Response 4: *I added in the introduction: “CDC hospitalization rates are significantly higher among those 65 years of age and older (by 4 times) and those over 85 years of age (by 10 times) compared to those under 65 years (18,19).”

*I also added in 2.2. Factors influencing the healthy gut microbiota: two recent studies about the gut microbiota composition in elderly patients with CDI. “Two interesting recent studies have addressed the gut microbiota composition in hospitalized elderly patients with CDI, showing lower microbial diversity, lower proportions of gut commensals with putative functions and a reduction of butyrate-production bacteria in CDI samples”

  1. Vakili, B.; Fateh, A.; Asadzadeh Aghdaei, H.; Sotoodehnejadnematalahi, F.; Siadat, S.D. Intestinal microbiota in elderly inpatients with Clostridioides difficile infection. IDR 2020, 13, 2723–2731, doi:10.2147/IDR.S262019.
  2. Milani, C.; Ticinesi, A.; Gerritsen, J.; Nouvenne, A.; Lugli, G.A.; Mancabelli, L.; Turroni, F.; Duranti, S.; Mangifesta, M.; Viappiani, A.; et al. Gut Microbiota composition and Clostridium difficile infection in hospitalized elderly individuals: a metagenomic study. Sci Rep 2016, 6, 25945, doi:10.1038/srep25945.

*I added in the conclusion: “The decrease of the diversity described in elderly gut microbiota (18), the decrease of some bacterial population (Ruminococcaceae, Bifidobacterium, Faecalibacterium) and the increase of some bacterial population (Proteobacteria, Bacteroidetes and Clostridium spp.) suggest why this population have a 10 times higher risk of developing CDI.”

Point 5: Line 111 “….TcdA and TcdB” If it is referred to protein, suggest use “toxin A and B”. If it is referred to gene, suggest use italic “tcdA, tcdB” 

Response 5: I changed with toxin A and B

Point 6: For 2.2. Immunity; line 101-118. What is the correlation between this paragraph with the topic “the microbiota change”?

Response 6: This paragraph was here to put a context and answer to the second objective: The secondary aim is to provide more information on why some people colonized with toxigenic C. difficile develop CDI and others show no signs of disease.

I reduced and merged this part with the introduction to avoid confusion.

Point 7: For the 3. Microbiota associated with asymptomatic colonization and CDI, line 149-239. These are the most important paragraph related to the topic. However the authors just list the findings from previous articles. As a review article suggest to provide more conclusions or suggestions for readers from these articles.

Response 7: I added in conclusions: “The gut microbiota will promote the development of the CDI. Through all the studies, the CDI has a gut microbiota footprint with the decrease and the increase of some bacteria. In this review, a lot of bacteria are singled out for giving an advantage or a disadvantage when developing CDI. Some of these bacteria have an impact on gut health. Faecalibacterium prausnitzii is considered as a species of the healthy gut microbiota. This bacterium is reduced in gut dysbiosis, in IBD, in obesity, in diabetes etc. [98]. Lachnospiraceae is protective against CDI [20]. C. scindens, member of Lachnospiraceae have a protective effect against CDI [20,32]. Amrane et al., (2018) showed that C. scindens is present in the feces when patient developing CDI [99]. Alistipes spp. indicated a controversial pathogenicity. On one hand, the bacteria have protective effects against liver fibrosis, cancer immunotherapy and cardiovascular disease [100]. On the other hand, this genus is associated with colorectal cancer and mental disease [100].

In this review, CDI can be associated with an increase or a decrease of A. municiphila and AC is associated with a decrease A. municiphila. The presence of this bacteria is positive against obesity, diabetes, cardiometabolic disease and low-grade inflammation [101]. It’s actually used to manage obesity [102]. In human intestinal organoids, C. difficile is capable of decreasing MUC2 production, but it is not responsible for altering host mucus oligosaccharide composition [31]. Furthermore, it has been reported that C. difficile is not capable of degrading mucin glycans, although coculture with mucin-degrading Akkermansia muciniphila, Bacteroides thetaiotaomicron and Ruminococcus torques allowed the pathogen to grow in media that lacked glucose but contained purified MUC2 [103]. When mucus is degraded by bacteria, oligosaccharides (GlcNAc, GalNAc, galactose, mannose and fucose) are salted out [103], and C. difficile is capable of using these oligosaccharides [31].

Enterobacteriaceae family is associated with the dysbiosis state [58,59]. Enterococcus spp. is a controversial bacterium [104]. It’s a commensal bacterium of intestinal flora, vagina, and mouth microbiota [104]. E. faecium and E. faecalis are used as probiotic [104,105] and Enterococcus spp. participated in meat and cheese [104,106] fermentation. Recently, E. faecalis and E. faecium are potentially pathogenic bacteria due to their ability to adapt in new environment [104,107]. Additionally, a resistance to vancomycin has emerged in this genus [104,107]. Romyasamit et al. (2020) exhibited that six E. faecalis strains have a probiotic effect and anti-C. difficile activity [108]. Klebsiella pneumonae is present in the mucus layer with C. difficile [109].”

Point 8: The same for Table 1 and Table 2. Suggest to provide more conclusions or suggestions for readers from these tables.

Response 8: I added in conclusions: “The gut microbiota will promote the development of the CDI. Through all the studies, the CDI has a gut microbiota footprint with the decrease and the increase of some bacteria. In this review, a lot of bacteria are singled out for giving an advantage or a disadvantage when developing CDI. Some of these bacteria have an impact on gut health. Faecalibacterium prausnitzii is considered as a species of the healthy gut microbiota. This bacterium is reduced in gut dysbiosis, in IBD, in obesity, in diabetes etc. [98]. Lachnospiraceae is protective against CDI [20]. C. scindens, member of Lachnospiraceae have a protective effect against CDI [20,32]. Amrane et al., (2018) showed that C. scindens is present in the feces when patient developing CDI [99]. Alistipes spp. indicated a controversial pathogenicity. On one hand, the bacteria have protective effects against liver fibrosis, cancer immunotherapy and cardiovascular disease [100]. On the other hand, this genus is associated with colorectal cancer and mental disease [100].

In this review, CDI can be associated with an increase or a decrease of A. municiphila and AC is associated with a decrease A. municiphila. The presence of this bacteria is positive against obesity, diabetes, cardiometabolic disease and low-grade inflammation [101]. It’s actually used to manage obesity [102]. In human intestinal organoids, C. difficile is capable of decreasing MUC2 production, but it is not responsible for altering host mucus oligosaccharide composition [31]. Furthermore, it has been reported that C. difficile is not capable of degrading mucin glycans, although coculture with mucin-degrading Akkermansia muciniphila, Bacteroides thetaiotaomicron and Ruminococcus torques allowed the pathogen to grow in media that lacked glucose but contained purified MUC2 [103]. When mucus is degraded by bacteria, oligosaccharides (GlcNAc, GalNAc, galactose, mannose and fucose) are salted out [103], and C. difficile is capable of using these oligosaccharides [31].

Enterobacteriaceae family is associated with the dysbiosis state [58,59]. Enterococcus spp. is a controversial bacterium [104]. It’s a commensal bacterium of intestinal flora, vagina, and mouth microbiota [104]. E. faecium and E. faecalis are used as probiotic [104,105] and Enterococcus spp. participated in meat and cheese [104,106] fermentation. Recently, E. faecalis and E. faecium are potentially pathogenic bacteria due to their ability to adapt in new environment [104,107]. Additionally, a resistance to vancomycin has emerged in this genus [104,107]. Romyasamit et al. (2020) exhibited that six E. faecalis strains have a probiotic effect and anti-C. difficile activity [108]. Klebsiella pneumonae is present in the mucus layer with C. difficile [109].”

Point 9: For the conclusion part, line 271-296. The authors just list again the findings from previous articles. The same for the conclusion part, suggest to provide more conclusions or suggestions for readers from these tables.

Response 9: I added in conclusions: “The gut microbiota will promote the development of the CDI. Through all the studies, the CDI has a gut microbiota footprint with the decrease and the increase of some bacteria. In this review, a lot of bacteria are singled out for giving an advantage or a disadvantage when developing CDI. Some of these bacteria have an impact on gut health. Faecalibacterium prausnitzii is considered as a species of the healthy gut microbiota. This bacterium is reduced in gut dysbiosis, in IBD, in obesity, in diabetes etc. [98]. Lachnospiraceae is protective against CDI [20]. C. scindens, member of Lachnospiraceae have a protective effect against CDI [20,32]. Amrane et al., (2018) showed that C. scindens is present in the feces when patient developing CDI [99]. Alistipes spp. indicated a controversial pathogenicity. On one hand, the bacteria have protective effects against liver fibrosis, cancer immunotherapy and cardiovascular disease [100]. On the other hand, this genus is associated with colorectal cancer and mental disease [100].

In this review, CDI can be associated with an increase or a decrease of A. municiphila and AC is associated with a decrease A. municiphila. The presence of this bacteria is positive against obesity, diabetes, cardiometabolic disease and low-grade inflammation [101]. It’s actually used to manage obesity [102]. In human intestinal organoids, C. difficile is capable of decreasing MUC2 production, but it is not responsible for altering host mucus oligosaccharide composition [31]. Furthermore, it has been reported that C. difficile is not capable of degrading mucin glycans, although coculture with mucin-degrading Akkermansia muciniphila, Bacteroides thetaiotaomicron and Ruminococcus torques allowed the pathogen to grow in media that lacked glucose but contained purified MUC2 [103]. When mucus is degraded by bacteria, oligosaccharides (GlcNAc, GalNAc, galactose, mannose and fucose) are salted out [103], and C. difficile is capable of using these oligosaccharides [31].

Enterobacteriaceae family is associated with the dysbiosis state [58,59]. Enterococcus spp. is a controversial bacterium [104]. It’s a commensal bacterium of intestinal flora, vagina, and mouth microbiota [104]. E. faecium and E. faecalis are used as probiotic [104,105] and Enterococcus spp. participated in meat and cheese [104,106] fermentation. Recently, E. faecalis and E. faecium are potentially pathogenic bacteria due to their ability to adapt in new environment [104,107]. Additionally, a resistance to vancomycin has emerged in this genus [104,107]. Romyasamit et al. (2020) exhibited that six E. faecalis strains have a probiotic effect and anti-C. difficile activity [108]. Klebsiella pneumonae is present in the mucus layer with C. difficile [109].”

Reviewer 4 Report

Martines et al. submitted their manuscript entitled “Does the microbiota influence the development of Clostridioides difficile infection?” to Pathogens. CDI is an interesting and actual topic, but I don’t think that the title of the manuscript is suitable. FMT and its obvious effect in recurrent infection document the importance of the microbiota on CDI without any question mark. Please, modify the title.

L39: CDI was introduced on line 33, bud AC was not introduced. The introduction in the abstract is not enough.

Figure 1. Here the column people older than 65 would be shown too. Does the advanced age mean more than 65? Please, clarify it. In the figure, a legend should be written which characteristics are depicted: Mean + S.D. or S.E.M., etc.

L105: It would be suitable at least briefly to describe that tight junction proteins join adjacent enterocytes (colonocytes) at their apical part to form a semipermeable intestinal barrier between the bacteria-rich lumen and the host’s body.

L111: It should be at least briefly mentioned differences between TcdA and TcdB.

L124: At the beginning, it would be suitable at least briefly to describe the role (bacteria translocation, digestion, penetration of toxins, …) of the mucins covering the epithelial layer of enterocytes (colonocytes).

I miss any mention dealing with animal models of CDI, e.g. Saul Tzipori’s group uses a gnotobiotic piglet model of CDI. Please, add a brief summary dealing at least with piglet and mouse models.

Author Response

Point 1: Martines et al. submitted their manuscript entitled “Does the microbiota influence the development of Clostridioides difficile infection?” to Pathogens. CDI is an interesting and actual topic, but I don’t think that the title of the manuscript is suitable. FMT and its obvious effect in recurrent infection document the importance of the microbiota on CDI without any question mark. Please, modify the title.

Response 1: I changed the title with : ”Gut microbiota composition associated with Clostridioides difficile colonization and infection

Point 2: L39: CDI was introduced on line 33, bud AC was not introduced. The introduction in the abstract is not enough.

Figure 1. Here the column people older than 65 would be shown too. Does the advanced age mean more than 65? Please, clarify it. In the figure, a legend should be written which characteristics are depicted: Mean + S.D. or S.E.M., etc.

Response 2: I reduced and merged this part with the introduction to avoid confusion. I added a section in the microbiota section to talk about the “2.2 factors influencing the gut microbiota”.

Lines 80- 82: I modified the legend and the category in the graph: “Figure 1. Box plot illustrating the mean relative proportion of C. difficile asymptomatic carriers in function of category of age. These data were found in articles that studied the prevalence of AC [4,20–27].

Point 3: L105: It would be suitable at least briefly to describe that tight junction proteins join adjacent enterocytes (colonocytes) at their apical part to form a semipermeable intestinal barrier between the bacteria-rich lumen and the host’s body.

Response 3: According to reviewer’ point, I reduced and merged this part with the introduction to avoid confusion. I added a section in the microbiota section to talk about the “2.2. factors influencing the gut microbiota”. Please note that I have focused my review on gut microbiota.

Point 4: L111: It should be at least briefly mentioned differences between TcdA and TcdB.

Response 4: According to other comments of reviewer, I reduced this part and focused on gut microbiota.

Point 5: L124: At the beginning, it would be suitable at least briefly to describe the role (bacteria translocation, digestion, penetration of toxins, …) of the mucins covering the epithelial layer of enterocytes (colonocytes).

Response 5: According to other comments by reviewer, I reduced this part and focused on gut microbiota. I added in the conclusion a discussion between modifying bacteria by C. difficile and their impact on health. And I discussed more about FMT and its implication with gut microbiota.

I miss any mention dealing with animal models of CDI, e.g. Saul Tzipori’s group uses a gnotobiotic piglet model of CDI. Please, add a brief summary dealing at least with piglet and mouse models.

Response 5: I added (Lines 102- 110) in the introduction the model of CDI. “Several studies have tried to understand the C. difficile pathogenesis in order of reduce the risks of development of the disease and find new therapeutic strategy. The animal experimentations using hamster have allowed to test the transplantation fecal efficiency, the use of non-toxigenic strain of C. difficile and use of monoclonal antibodies against toxin A and toxin B [33–35]. The piglet model of CDI is representative of the key characteristics of human CDI and helped to understand the virulence and new treatment [36]. Then, C. difficile studies use different methods in vitro in order to limit using animal experimentation: feces cultures [37], continuous culture model [38,39], triple-stage chemostat human gut model [40,41], Tim-2 model [42]»

Round 2

Reviewer 2 Report

The authors have taken sufficient efforts to improvise this manuscript. I am happy with the way this manuscript has evolved.

Reviewer 3 Report

had been revised as suggestions

Reviewer 4 Report

I have no additional comments.